# An Efficient Continuous Flow Synthesis for the Preparation of *N*-Arylhydroxylamines: Via a DMAP-Mediated Hydrogenation Process

**DOI:** 10.3390/molecules28072968

**Published:** 2023-03-27

**Authors:** Jianli Chen, Xinyu Lin, Feng Xu, Kejie Chai, Minna Ren, Zhiqun Yu, Weike Su, Fengfan Liu

**Affiliations:** 1College of New Materials Engineering, Jiaxing Nanhu University, Jiaxing 314000, China; 2National Engineering Research Center for Process Development of Active Pharmaceutical Ingredients, Collaborative Innovation Center of Yangtze River Delta Region Green Pharmaceuticals, Zhejiang University of Technology, Hangzhou 310014, China; 3Raybow (Hangzhou) Pharmaceutical Co., Ltd., Hangzhou 310014, China

**Keywords:** *N*-arylhydroxylamine, 4-(dimethylamino) pyridine, continuous flow, catalytic activity, selectivity

## Abstract

The selective hydrogenation of nitroarenes to *N*-arylhydroxylamines is an important synthetic process in the chemical industry. It is commonly accomplished by using heterogeneous catalytic systems that contain inhibitors, such as DMSO. Herein, DMAP has been identified as a unique additive for increasing hydrogenation activity and product selectivity (up to >99%) under mild conditions in the Pt/C-catalyzed process. Continuous-flow technology has been explored as an efficient approach toward achieving the selective hydrogenation of nitroarenes to *N*-arylhydroxylamines. The present flow protocol was applied for a vast substrate scope and was found to be compatible with a wide range of functional groups, such as electron-donating groups, carbonyl, and various halogens. Further studies were attempted to show that the improvement in the catalytic activity and selectivity benefited from the dual functions of DMAP; namely, the heterolytic H_2_ cleavage and competitive adsorption.

## 1. Introduction

*N*-Arylhydroxylamines (*N*-AHA) are versatile organic molecules used as intermediates to synthesize industrially valuable fine chemicals, bioactive drugs, and polymerization inhibitors [1,2,3,4,5,6]. Among the reported protocols for obtaining *N*-AHA, the selective reduction of nitroarenes is the most convenient, which involves a biocatalytic reduction by baker’s yeast or plant cell [7,8], the use of Fe or Zn in stoichiometric quantity [9,10,11], and homogeneous or heterogeneous metal catalysts (with Pt, Ni, Pd, Rh, Ir, Ag, Au) with hydrazine or H_2_ [12,13,14,15,16,17,18,19]. At present, lots of attention is focused on the development of a heterogeneous metal catalytic system because of its great stability, reusability, and easy isolation.

As the reduction of nitroarenes is a complex reaction network consisting of nitrosobenzene, aromatic amine, hydroxylamine, hydrazine, azoxybenzene, and azobenzene, the preparation of *N*-AHA with high selectivity has always been a challenge (Figure 1) [20,21,22]. In the catalytic reduction involving transition metals, palladium, and particularly platinum, are preferred due to their great activity and *N*-AHA selectivity. However, commercially available Pt or Pd catalysts supported solely on carbon, silicon, or other supports still possess the issue of low *N*-AHA selectivity, owing to the formation of significant amounts of aromatic amines in the reaction process [23,24,25,26]. The control of the reaction conditions, such as the H_2_ pressure, temperature, and catalyst usage is conducive to retarding the formation of aromatic amines and increasing the *N*-AHA selectivity, but it is not effective enough. Lowering the adsorption tendency of *N*-AHA on the catalyst surface is a powerful strategy to enhance the selectivity, which can prevent *N*-AHA from being converted into azoxybenzene or aromatic amine. For example, Zheng et al. significantly increased the selectivity of *N*-AHA by introducing the modifier ethylenediamine (EDA) in the preparation of ultrathin platinum nanowires (Pt NWs) [27]. Mechanistic studies have indicated that the electron donation of ethylenediamine enriches the electron of the Pt NWs surface, which hinders the adsorption of *N*-AHA. Although the EDA-Pt NWs can significantly improve the selectivity of *N*-AHA, it is still difficult to obtain excellent selectivity for *N*-AHA substituted by electron-donating groups. The mechanism is also applicable to Pd/C_6_F_13_-Cell, as prepared by Cai et al [28]. Beyond that, in an effect to increase the selectivity of *N*-AHA, it is a common practice to add chemical reagents to form strongly competitive adsorption with *N*-AHA in the reaction process. It is well-known that the addition of dimethylsulfoxide (DMSO) into a Pt-catalyzed system can significantly increase the selectivity of *N*-AHA, but unfortunately it also decreases the reaction activity [24,29]. Takenaka et al. found that the addition of a nitrogen-containing base can improve the hydrogenation reaction activity, which is used to compensate for the low catalyst activity caused by DMSO [24]. This practice undoubtedly makes post-processing more cumbersome. Meanwhile, the inhibition of DMSO on the catalyst activity is not conducive to the recovery and reuse of the catalyst.

In the cases of reducing nitroarene to *N*-AHA, sodium borohydride and hydrazine hydrate are the preferred reducing agents because they can provide better selectivity [12,13,14,15,16]. In contrast, H_2_, an environment-friendly and economical reductant, was carefully selected. Although the batch process is widely used in organic synthesis due to its versatility and flexible production planning and scheduling, its application in heterogeneous hydrogenation is limited by the poor gas/liquid/solid multiphase mixing performance and the high flammability of H_2_. Our group has been exploring the protocols to achieve the efficient preparation of fine chemicals through continuous-flow technology [30,31,32,33]. Continuous-flow technology has unique advantages in terms of the process enhancement, such as safer operating environment, excellent mass and heat transfer, accurate control over the process parameters, low back-mixing, solid catalyst immobilization, etc. [34,35,36,37]. To date, there have been many reports on the application of continuous-flow technology in gas/liquid/solid heterogeneous hydrogenation [38,39,40]. However, to the best of our knowledge, there are only a few cases of the hydrogenation of nitroaromatics to *N*-AHA using continuous-flow technology [41,42]. The continuous-flow technology has shown great control over the consecutive reaction, but the selectivity of the *N*-AHA obtained is not satisfactory in the micro-packed bed (µPBR). Dounay et al. found that the addition of 1,10-phenanthroline (1,10-phen) in the continuous-flow system with 5 wt.% Pt/C can increase the yield of *N*-AHA [41]. Unfortunately, this protocol showed poor functional group tolerance (0–83% yield). Meanwhile, the inhibition of 1,10-phen on the catalyst activity cannot be ignored. In our previous study, we found that *N*-AHA could not be observed at low conversion when the µPBR was filled with Raney Ni for the hydrogenation of nitroarenes [42]. When DMSO-modified Raney Ni was filled in µPBR, the formation of *N*-AHA with high selectivity can be observed. However, the DMSO-modified Raney Ni was seriously deactivated and had poor functional group tolerance (39–99% selectivity). 

In this paper, we report a continuous-flow system for the highly efficient preparation of *N*-AHA by adding only one additive, 4-(dimethylamino) pyridine (DMAP), which avoided the high cost of preparing new catalysts and the loss of catalyst activity caused by the inhibitors. The reaction conditions, such as the liquid flow rate and the temperature, will be investigated to show the significant effect of DMAP in promoting the formation of *N*-AHA and retarding the formation of over-reduction products. To the best of our knowledge, this is the first report of using only one additive to improve both the selectivity and activity in heterogeneous catalysis. Moreover, a DFT calculation will be attempted to reveal the mechanism of the dual functions of a single additive.

## 2. Results and Discussion 

### 2.1. Reaction Profile in Autoclave

*o*-Chloronitrobenzene (**1a**) was selected as the model substrate for investigation. To start with, the catalytic performance of 5 wt.% Pt/C was evaluated in autoclave without the influence of external diffusion (Figure 1). At 60 °C, the highest amount of **1b** reaches 50%, accompanied by a large number of *o*-chloroaniline (**1c**) (Figure 1a). With the extension of the reaction time, **1b** will eventually be fully converted into **1c**, but at the same time, the formation of aniline (**1d**), a by-product of dehalogenation, can be observed. Even when the temperature drops to 30 °C, the maximum amount of **1b** can only reach about 70%, and a longer reaction time is required (Figure 1b). There is no doubt that using 5 wt.% Pt/C as a catalyst for catalytic hydrogenation in an autoclave cannot obtain the desired selectivity towards **1b** and has the problem of an uncontrollable process. Subsequently, 5 wt.% Pt/C was filled in the µPBR to further explore the continuous flow conditions.

### 2.2. Optimization of the Experimental Conditions in Flow

Solvents are known to have a significant effect on the activity and selectivity of catalytic hydrogenation, which includes the solubility of H_2_, the interaction of the solvent with chemical components, and the competitive adsorption of the solvent, etc. [15,43]. As different solvents may lead to different reaction routes, it is necessary to evaluate the solvent before investigating the additives. It was be observed that, with the exception of 1,4-dioxane, the polarity of the solvent has a positive correlation with the reactivity (Figure 2). It is supposed that the reason for the abnormal outcome of using 1,4-dioxane as the solvent is that two oxygen atoms with high electron density easily occupy the active sites and compete with the reaction components for adsorption. The polar protic solvent methanol (MeOH) showed the highest activity, which significantly improves the hydrogenation activity, leading to trace of **1b** and 27% of **1d**. Ethyl acetate (EA), dichloromethane (DCM), and tetrahydrofuran (THF) provided similar conversions, where THF can afford **1b** with the selectivity of 77% at a conversion of 82%. However, ~1% **1d** was observed in the above three organic solvents, which is probably due to the extremely high catalyst/substrate in the µPBR, leading to more active sites being adsorbed by **1c**. Notably, a significant amount of azoxy intermediates (**1e**) were observed in the acetone and ether, which may be caused by the condensation of nitrosobenzene and *N*-AHA during the reaction. Consistent with the reports in the literature, THF is the preferred solvent for the preparation of *N*-AHA [15,44]. Considering that it is more likely to produce **1c** under near-full conversion, even if THF is used as a solvent in the continuous flow, the maximum content of **1b** will be similar to that in the batch under full conversion, and the appearance of **1d** cannot be ignored.

The selected additives are common inhibitors, such as DMSO, and promoters, such as amines, in the literature, as well as their analogues. As the addition of some amines at 25 °C requires a high liquid flow rate to clearly observe the effect on the reaction, which will lead to greater resistance in the µPBR, the effect of the additives was investigated by appropriately increasing the liquid flow rate at 15 °C. No dehalogenation by-products were observed in the reaction system after the temperature decreased (Table 1, entry 1). When the liquid flow was 0.5 mL·min^−1^, it was observed that the addition of *N*,*N*,*N′*,*N’*-tetramethylethylenediamine (TMEDA), ammonia, and triethylamine (TEA) did not increase the selectivity towards **1b** at full conversion (Table 1, entries 2, 4, 6), which may be the result of the long residence time. As the liquid flow is increased to 0.9 mL·min^−1^, a certain amount of **1c** affects the selectivity of **1b** (92–95%) at high conversion (88–94%) (Table 1, entries 3, 5, 7). Additives with high alkalinity, such as TMEDA, ammonia, and TEA, can significantly increase the hydrogenation activity, but not the selectivity, which can be explained by their poor adsorption capacity not being enough to form strong competitive adsorption with **1b**. Consistent with what has been reported in the literature [24,41], DMSO and 1,10-phen inhibit the activity of the catalyst (Table 1, entries 8–9), mainly by occupying the catalytic active site to block the adsorption of **1a** and **1b**, resulting in a slower reaction rate, which is conducive to the build-up of **1b**. In particular, the addition of pyridine increased the hydrogenation rate of **1a** and the selectivity of **1b** (Table 1, entry 11). We speculate that the mechanism of the hydrogenation of **1a** promoted by pyridine is the same as that of amines, which is related to its alkalinity. Inspired by this, DMAP, a pyridine derivative with higher alkalinity, was selected as an additive for investigation (Table 1, entry 10). It was found that DMAP greatly increased the hydrogenation rate of **1a** and still provided a selectivity of >99%. Compounds containing other heteroatoms, such as cyclohexylmethyldimethoxysilane (CHMMS), were also investigated as additives, but they did not provide satisfactory outcomes (Table 1, entry 12). DMAP was chosen as an ideal additive to investigate the effect of the reaction parameters.

As low temperature is conducive to the formation of *N*-AHA, the temperature is set at 25 °C again to avoid the interference of the temperature. When the equivalent of DMAP increases, the conversion and selectivity will be increased until >99% (Figure 3a). However, the conversion will remain stable when the equivalent is increased from the minimum 0.05 eq. in batch (Appendix A). The difference between the two results is attributed to the ratio of catalyst to substrate in the batch, which is far less than that in the µPBR. As there are still enough active sites in the µPBR to be occupied by *N*-AHA when 0.08 eq. DMAP is added, the selectivity is not satisfactory at full conversion. For the continuous-flow hydrogenation process, the optimal equivalent of DMAP is 0.1.

The reaction temperature was varied between 15 °C and 45 °C to study the effect on the conversion and selectivity (Figure 3b). When the temperature is 15 °C, full conversion cannot be obtained. With a further increase in the temperature, the selectivity of **1b** will decrease under the full conversion. As the activation energy of the hydrogenation of nitroarenes to *N*-AHA is less than that of the hydrogenation of *N*-AHA to anilines, a low temperature is often selected to obtain *N*-AHA with high selectivity [26]. However, it was found that the selectivity of **1b** reached 95% even when the temperature rose to 45 °C, indicating that the addition of DMAP significantly inhibited the transformation of **1b** to **1c**. Furthermore, no dehalogenation by-products were observed, even at 45 °C. For the continuous-flow hydrogenation process, the optimal reaction temperature is 25 °C.

The liquid flow rate is closely related to the residence time; that is, a higher liquid flow rate leads to a longer residence time, and a lower liquid flow rate leads to a shorter residence time. The liquid flow rate was varied between 0.25 mL·min^−1^ and 1.25 mL·min^−1^ to study the effect on the conversion and selectivity (Figure 3c). An increase in the liquid flow rate led to a significant decrease in the conversion. However, when the liquid flow rate dropped to 0.25 mL·min^−1^, the selectivity of **1b** still remained at 98%, which also reflected the strong inhibition of DMAP on the transformation of **1b** to **1c**.

### 2.3. Applicability of the Continuous Flow System

With these optimized reaction conditions in hand, we have explored the substrate scope of this continuous flow system (Figure 2). However, 0.1 eq. DMAP can only guarantee that the hydrogenation of **1a**, *o*-bromonitrobenzene (**4a**), *o*-iodonitrobenzene (**5a**), nitrobenzene (**6a**), methyl 4-nitrobenzoate (**7a**), and *p*-nitroacetophenone (**8a**) can be conducted with excellent conversion and selectivity. After optimizing the conditions, we found that the efficient hydrogenation of *m*-chloronitrobenzene (**2a**), *p*-chloronitrobenzene (**3a**), *o*-methylnitrobenzene (**9a**), and 1-(4-chlorophenyl)-3-((2-nitro- benzyl)oxy)-1H-pyrazole (**10a**) can be achieved by appropriately increasing the equivalent of DMAP. Notably, no dehalogenation by-products were tested in the hydrogenation process of **1–5a** and **10a**. The conversion of **10a**, an important intermediate for the synthesis of a new broad-spectrum bactericide, is only 94%, which is speculated to be caused by the poor solubility of the raw material in THF. 

Based on the above results, the electronic effect and steric effect are considered to be important factors affecting the selectivity of *N*-AHA. Nitroarenes containing electron-donating groups are more likely to be converted into anilines than those containing electron-withdrawing groups, resulting in the low selectivity of *N*-AHA [27,41,45]. The electron-donating group (**9a**, **10a**) could enrich the electrons of -NHOH, leading to the tendency of -NHOH to adsorb on the catalyst surface and convert to -NH_2_. The steric effect depends on the positions of the substituents [17,41]. In the cases of chloronitrobenzene (**1a**–**3a**) hydrogenation, **1a** and **1b** have the largest steric hindrance, while that of **3a** and **3b** are the opposite. Thus, **3b** is preferable to be adsorbed on the catalyst and converted into aromatic amine. Increasing the equivalent of DMAP can improve the undesirable selectivity caused by the electronic effect and steric effect, which is attributed to the increase in the DMAP coverage on the catalyst surface [46], resulting in more difficulty for *N*-AHA to adsorb on the catalyst surface. 

In addition, we attempted to change the steric effect and electronic effect based on the structure of DMAP to explore more effective additives. The structural analogues, 2-dimethylaminopyridine (2-DMAP) and 4-(dimethylamino) triphenylphosphine (4-DTP), were used as additives to investigate the selective hydrogenation of **3a** and **9a** (Table 2). Compared with DMAP, the result of adding 2-DMAP was undesirable; that is, the conversion and selectivity increased slowly with the increase in equivalent (Table 2, entries 1–3). It is supposed that 2-DMAP has higher steric hindrance, which makes it more difficult to adsorb on the catalyst surface than DMAP. The hydrogenation of **3a** and **9a** was severely inhibited by 4-DTP, which is attributed to the stronger adsorption capacity of 4-DTP than that of DMAP, preventing the adsorption of nitroarenes (Table 2, entries 4–5). DMAP is the most ideal additive in terms of the reaction efficiency, cost, and tolerance for various functional groups.

After successfully evaluating the generality of the scope with various nitroarenes, we became interested in verifying the effectiveness of the catalyst system in our flow setup. Accordingly, we conducted the hydrogenation for 20 hours under the optimal conditions using **1a** (Appendix A). Notably, the catalytic activity in the flow process was maintained, even after 20 h, with >99% conversion and >99% selectivity.

### 2.4. Mechanisms of Reaction

On the premise of the elimination of the external diffusion limitation, the comparison of the reaction profiles clearly shows that the addition of DMAP accelerates the conversion of 1a and retards the formation of 1c (Figure 1 and Figure 4). The results from the blank test revealed that the reaction could only proceed when the catalyst, hydrogen, and DMAP were all present in the system (Appendix A). At present, there is no relevant report on the mechanism of additives in the reduction of nitroarenes. The research on the mechanism of the increase in the reaction rate or selectivity caused by additives is more concentrated in the selective hydrogenation of alkynes [47,48,49]. However, it has not been reported in catalytic hydrogenation that both the selectivity and activity can be increased through only one additive.

To date, many papers have systematically reported the reasons for the increase in the semi-hydrogenation selectivity caused by the addition of non-reactive compounds [46,47,48]. Kaeffer et al. demonstrated that the adsorption rate constant of ligands (PCy_3_ or IMes) added is similar to that of alkynes and is several orders of magnitude higher than that of alkenes, which is conducive to competitive adsorption between ligands and alkenes [50]. Shevchenko et al. also conducted similar studies [46]. They believed that the adsorption energy of capping ligands at the catalytic surface is higher than that of alkynes, which will lower the catalytic activity, resulting in the low conversion of alkynes. Based on the analysis of the experimental results caused by the addition of 2-DMAP and 4-DTP (Table 2), we suggest that the mechanism of competitive adsorption without affecting the conversion of nitroarene is that the adsorption energy of the additives is close to that of nitroarenes, so that they can cover the catalyst surface at the initial stage of the reaction; during the reaction, because of the low adsorption energy of the generated *N*-AHA, it is hard for *N*-AHA to make contact with the active site when competitive adsorption has been generated. 

Bearing in mind that amines are used as ligands or additives in homogeneous or heterogeneous catalysis to promote the heterolytic cleavage of H_2_ to give metal hydride species [49,50,51,52], we suggest that another crucial role of DMAP is to facilitate the heterolytic H_2_ activation at the Pt/C. Namely, the nitrogen atom of DMAP can serve as a basic ligand to form a frustrated Lewis pair (FLP) with the active site of Pt/C, which can promote the heterolytic H_2_ cleavage. This type of H_2_ activation with DMAP as an additive was previously described for NH_3_ and piperazine [49,53]. However, the two nitrogen atoms of DMAP in different chemical environments require us to distinguish their functions. In the latter part, N1 and N2 represent the nitrogen atom of the pyridine ring and the nitrogen of the dimethylamino group, respectively.

At the initial calculation stage, the adsorption energy of N1 and N2 on the Pt (111) surface was calculated (Figure 5A,B). The calculated adsorption energy of N1 and N2 is -0.47 eV and −0.29 eV, respectively, indicating that the N1 adsorb on Pt more strongly. Moreover, it was observed that there is a significant transfer of electrons when N1 adsorbs on the Pt surface, while the opposite is true for N2 (Figure 5C,D). As a result, adsorption is considered as the interaction between N1 and the Pt surface, which is consistent with the fact that N1 is more capable of complexing with metals, as reported in the literature [54,55].

As the DMAP adsorption is the result of the lone pair electrons of N1 filling into the *d* orbits of Pt, N1 cannot form FLP with the active site of the Pt after adsorption. The formation mechanisms of five FLPs are considered, as follows (Figure 6): (A) The N2 of the adsorbed DMAP formed FLP with the active site of the Pt/C (HED-1). (B) Under the mechanism of A, the effect of the electron transfer caused by DMAP adsorption is considered (HED-2). (C) The N2 of the free DMAP formed FLP with the metal active center of the Pt/C (HED-3). (D) Under the mechanism of C, the effect of the electron transfer caused by DMAP adsorption is considered (HED-4). (E) Considering the electron transfer caused by the adsorption of DMAP, the N1 of the free DMAP formed FLP with the metal active center of the Pt/C (HED-5). 

We have computed the adsorption (reaction) energies of H_2_ for the above five mechanisms and the homogeneous dissociation (HOD) mechanism of H_2_ (Figure 7 and Appendix A). In the absence of DMAP, the homogeneous dissociation of H_2_ occurs on the Pt surface and forms two H atoms with negative charges (Appendix A). The adsorption energy calculated by HED-1 is −0.58 eV, which is significantly higher than the adsorption energy calculated by HOD. After heterolytic H_2_ cleavage, the H atom close to the catalyst surface is negatively charged, while the H atom close to the N2 is opposite (Appendix A). As the adsorption of DMAP causes N2 to be far from the catalyst surface, FLP can still be formed to promote H_2_ activation. Furthermore, the electron transfer between the adsorbed DMAP and Pt has little effect on the H_2_ activation; that is, the adsorption energy of HED-2 is only 0.03 eV higher than that of HED-1. It can be seen from the above that the adsorption of DMAP enriches the positive charge of the Pt to facilitate the heterolytic H_2_ activation, but it is not the key factor. High dissociation adsorption energy (−0.77 eV) was calculated following HED-3, which is supposed to be the result of the closer distance between N2 and the active site, leading to the more effective formation of FLP. When the influence of the electron transfer between the adsorbed DMAP and Pt on H_2_ activation is considered, the adsorption energy of HED-4 is 0.06 eV higher than that of HED-3. The adsorption energy calculated for HED-5 is only -0.52 eV, which is supposed to be the result of the weak alkalinity of N1. According to the DFT calculations, the magnitude of the adsorption energy for H_2_ increases as HOD < HED-5 < HED-1 < HED-2 < HED-3 < HED-4. HED-4 is considered to be the possible mechanism for the heterolytic cleavage of H_2_. 

In addition, the effects of DMAP and TEA with different equivalents on the reaction were further compared in batch (Appendix A). In previous studies, it has been observed that the reaction rate remains stable when the amount of DMAP added increases from 0.05 eq. (Appendix A). However, the equivalent of TEA increases from 0.05 eq. to 0.1 eq., leading to a further increase in the reaction rate. The difference in the results is supposed to be caused by the difference in the adsorption capacity and alkalinity of the additives. As the adsorption capacity of TEA is less than DMAP, more TEA distribution around the catalyst surface with the increase in equivalent is beneficial to the formation of FLP. The strong adsorption of DMAP can not only ensure sufficient FLP formation at a low equivalent, but can also increase the selectivity by competitive adsorption with *N*-AHA. As the higher alkalinity of TEA is conducive to heterolytic H_2_ activation [49], the addition of 0.1 eq. TEA showed a greater effect on the reaction rate than DMAP. Based on the above investigation, we conjecture that the alkalinity and the adsorption capacity of additives are the key factors affecting the heterolytic H_2_ activation.

On the basis of the experimental and theoretical results described above, we propose the mechanism for the hydrogenation of nitroarenes by Pt/C in the presence of DMAP, which is possible in the presence of DMAP by reason of the heterolytic activation of H_2_ via frustrated Lewis pairs and competitive adsorption (Figure 8).

## 3. Experimental Section

### 3.1. Materials and Methods

All of the purchased chemicals were used without further purification, unless indicated otherwise. The Nuclear Magnetic Resonance Spectroscopy (NMR) data recorded for the purified product were obtained using the BRUKER AVANCE III HD system (Hanau, Germany). The chemical composition of the reaction mixture was analyzed with High-Performance Liquid Chromatography (HPLC) (Agilent 1100 series equipped with a C18 chromatographic column). The following experimental parameters were used during the sample analysis: HPLC-grade acetonitrile and water were used to prepare the mobile phase. Gradient percentages of acetonitrile were used as the eluent: 25%, 3 min; 25–80%, 25 min; 80%, 5 min; and 25%, 7 min. The detection wavelength was 254 nm. The normalized peak areas obtained from the HPLC chromatograms were used for calculating the conversion of nitroarenes (**NB**), X, selectivity of ***N*-AHA**, S.
(1)$$X\;=\;\frac{\left[area\;of\;NB\;in\;starting\;solution\right]\;-\;[area\;of\;NB\;after\;reaction]}{\left[area\;of\;NB\;in\;starting\;solution\right]}$$
(2)$$S\;=\;\frac{\left[area\;of\;N\;-\;AHA\right]}{\left[area\;of\;N\;-\;AHA\right]\;+\;[area\;of\;all\;by\;-\;products]}$$

### 3.2. µPBR Continuous-Flow Reactor Setup

The µPBR was placed vertically in the water bath for temperature control (Figure 9). The flow direction was from the bottom to the top. The liquid flow was maintained using a dual-piston HPLC pump (LC-500PC). The gas and liquid were mixed in a T-shape mixer before it entered the µPBR (SS–98). All of the tubes were made of stainless steel (outer diameter: 1/16 inch). A manual back-pressure regulator was placed at the outlet of the reactor to control the system pressures.

The μPBR was made of a stainless steel tube with an inner diameter of 3.0 mm and a length of 50 mm. Next, 0.1 g of the catalyst was filled in the middle of the μPBR, and both ends were filled with 0.3 × 2 g of quartz sand and 0.05 × 2 g of cotton (Figure 10). The catalyst was filled as tight as possible. Due to its small size, the temperature in μPBR was thought to be consistent with the water bath.

### 3.3. A Typical Procedure for the Hydrogenation of Nitroarenes in Flow

The filled µPBR was first flushed with methanol at a rate of 0.5 mL·min^−1^ over a period of 30 min after being put into the continuous flow device. The solution of o-chloronitrobenzene (0.1 M) and DMAP (0.01 M) in THF was pumped into the flow system installed with a catalyst cartridge (5 wt.% Pt/C) at 30 °C under the H_2_ pressure of 0.6 MPa. The flow rate was maintained at 0.5 mL·min^−1^. After purging the solution for 30 min, the sample was collected 3 times. The sampling interval was 20 min. Following the completion of the experiments, the water bath was removed first, and the backpressure was released into the atmosphere. Following the process of depressurization, the µPBR was drained using H_2_. Following this, EtOAc was introduced into the µPBR. The EtOAc (50 mL) was flown before the reactor was sealed with this solvent for storage.

### 3.4. A Typical Procedure for the Hydrogenation of Nitroarenes in Batch

o-Chloronitrobenzene (1.0 g) was added in a high-pressure autoclave (40 mL) and 5 wt.% Pt/C (10 mg), DMAP (0.078 g), THF (20 mL), and magnetic stir bar were added to it, and the system was sealed. A vacuum–nitrogen cycle was conducted thrice to replace the air inside the reactor with nitrogen. Initially, the system was filled with hydrogen (0.5 MPa) at 20–30 °C, following which the pressure was released into the atmospheric. The cycle was repeated thrice. Finally, the system was filled with hydrogen (1.0 MPa), following which the system was sealed. The reaction mixture was stirred at a set speed after the temperature reached the set temperature. The reaction mixture (150 μL) was collected and filtered. The solution (100 μL) was collected from it, and 500 μL of acetonitrile was added to the sample. The diluted reaction solution (5 μL) was injected into the HPLC system for analysis.

### 3.5. Computational Details

First-principles calculations were carried out using the density functional theory (DFT) with a generalized gradient approximation (GGA) of Perdew-Burke-Ernzerhof (PBE) implemented in the Vienna Ab-Initio Simulation Package (VASP) [56,57]. The valence electronic states were expanded on the basis of plane waves with the core-valence interaction, represented using the projector augmented plane wave (PAW) approach and a cutoff of 520 eV [58]. A Γ-centered k-mesh of 1 × 1 × 1 was used for the surface calculations. Convergence is achieved when the forces acting on the ions become smaller than 0.02 eV/Å. 

## 4. Conclusions

In summary, we have described an exceedingly efficient approach for the mild hydrogenation of a wide range of nitroarenes to the corresponding N-AHA using Pt/C combined with DMAP based on continuous-flow technology. It is of practical significance that various synthetically useful functional groups, including the electron-donating groups, carbonyls, and halogens, can be compatible during nitroarene hydrogenation. Furthermore, the combined experimental/computational study attempted to explain that the dual role of DMAP in enhancing the selectivity and activity follows the heterolytic activation of H_2_ and the competitive adsorption between the DMAP and reaction components. A judicious selection of additives with appropriate adsorption capacity and basicity can lead to a remarkable enhancement in the catalytic selectivity and activity for selective hydrogenation. We expect the general findings and methodology of this study to become an efficient tool in the rational selection of additives in the selective hydrogenation with heterogeneous catalyst. We will explore the application of the DMAP-mediated hydrogenation process in a multi-step continuous synthesis.

## Data Availability

The data presented in this study are available in the article and Appendix A.

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
