# Peer review of "An Efficient Continuous Flow Synthesis for the Preparation of N-Arylhydroxylamines: Via a DMAP-Mediated Hydrogenation Process"

_molecules, 2023, doi:10.3390/molecules28072968_

Round 1

Reviewer 1 Report

Chen, Liu, and co-workers developed a highly selective and efficient method for the preparation of N-Arylhydroxylamines through hydrogenation of nitroarenes using Pt/C as catalyst with 4-dimethylaminopyridine (DMAP) as catalytic additive under continuous flow synthesis. The function of DMAP was proposed to block the adsorption of N-Arylhydroxylamines on the catalyst surface to prevent further reduction as well as facilitate the heterolytic hydrogen activation. The reaction is very selective and showed satisfied substrate scope. The work is comprehensive, and the manuscript is well-written and very well-organized. I recommend its publication in Molecules after addressing the following aspects.

1.       In Figure 3b, the authors did not show the data of 1d, does it mean the byproduct 1d is not formed or observed?

2.       In Table 1, the authors showed the additive effect of the reaction. For entry 8 and entry 9, both 1a(%) and 1b(%) are the same. I assume 1a(%) is the remaining 1a after reaction and 1b(%) is the yield of 1b. If it is the case, for both entries 1a(%)+1b(%) is over 100%. Also, for entry 11, the data is wired as both 1a(%) and 1b(%) is 44 and what is the rest 12% converted to? The authors should double-check the data and interpret the data clearly.

3.       The authors proposed the mechanism in Figure 10, The arrow push for the hydrogenation of nitroarene and aryl hydroxylamines needs to be redrawn.

4.       Based on the substrate scope, substrate 10b showed decreased conversion and selectivity compared to other substrates. Have the authors tested other complex substrates to check the compatibility under the condition?

Author Response

Chen, Liu, and co-workers developed a highly selective and efficient method for the preparation of N-Arylhydroxylamines through hydrogenation of nitroarenes using Pt/C as catalyst with 4-dimethylaminopyridine (DMAP) as catalytic additive under continuous flow synthesis. The function of DMAP was proposed to block the adsorption of N-Arylhydroxylamines on the catalyst surface to prevent further reduction as well as facilitate the heterolytic hydrogen activation. The reaction is very selective and showed satisfied substrate scope. The work is comprehensive, and the manuscript is well-written and very well-organized. I recommend its publication in Molecules after addressing the following aspects.

Thanks for the reviewer’s comments for our work. And the detailed responses for the concerns are provided as follows.

  1. In Figure 3b, the authors did not show the data of 1d, does it mean the byproduct 1d is not formed or observed?

Answer: Yes, dehalogenation by-product 1d was not observed in the batch reactor at 303K.

  1. In Table 1, the authors showed the additive effect of the reaction. For entry 8 and entry 9, both 1a(%) and 1b(%) are the same. I assume 1a(%) is the remaining 1aafter reaction and 1b(%) is the yield of 1b. If it is the case, for both entries 1a(%)+1b(%) is over 100%. Also, for entry 11, the data is wired as both 1a(%) and 1b(%) is 44 and what is the rest 12% converted to? The authors should double-check the data and interpret the data clearly.

Answer: We sincerely thank the reviewer for the constructive suggestion. We are very sorry for not checking again after writing the data. The corresponding modifications have been marked in Table 1.

  1. The authors proposed the mechanism in Figure 10, The arrow push for the hydrogenation of nitroarene and aryl hydroxylamines needs to be redrawn.

 Answer: Thanks for the reviewer’s suggestion. The drawing of the mechanism (Figure 10) refers to the mechanism proposed by Song et al (Chemical Engineering Journal, 2022, 429, 132224). In Figure 10, we use dotted lines to represent the binding of Hδ+ and Hδ- to unsaturated functional groups, which do not represent the direction of charge transfer. We think that using dotted lines is a suitable option.

  1. Based on the substrate scope, substrate 10b showed decreased conversion and selectivity compared to other substrates. Have the authors tested other complex substrates to check the compatibility under the condition?

Answer: We have not tested other complex compounds under the condition.

Reviewer 2 Report

The submitted manuscript reports a detailed protocol for selective hydrogenation of a wide range of nitroarenes to N-arylhydroxylamines, useful industrially synthesized organic intermediates. The authors have done an impressive amount of work to optimize the process conditions in order to avoid the formation of by-products and to obtain target compounds with yields close to quantitative. Proposed protocol of hydrogenation using Pt/C combined with DMAP as additive based on continuous-flow technology may be of interest to specialists in applied organic chemistry. The manuscript is well written. I recommend publication of this work with the minor corrections to improve the article:

Lines 11, 17, “N-Arylhydroxylamines” should be replaced by “N-arylhydroxylamines”

Lines 142, 163, “o-chloronitrobenzene” should be replaced by “o-Chloronitrobenzene”

Figure 4: “Ethyl” should be replaced by Ether

Line 282. “The +R effect of the electron-donating group (9a, 10a) could enrich the electrons of -NHOH, leading to the…” Electron-donating alkyl groups such as methyl (9a) have only positive inductive effect (+I effect). What kind of electronic effect do you mean?

Author Response

The submitted manuscript reports a detailed protocol for selective hydrogenation of a wide range of nitroarenes to N-arylhydroxylamines, useful industrially synthesized organic intermediates. The authors have done an impressive amount of work to optimize the process conditions in order to avoid the formation of by-products and to obtain target compounds with yields close to quantitative. Proposed protocol of hydrogenation using Pt/C combined with DMAP as additive based on continuous-flow technology may be of interest to specialists in applied organic chemistry. The manuscript is well written. I recommend publication of this work with the minor corrections to improve the article:

Lines 11, 17, “N-Arylhydroxylamines” should be replaced by “N-arylhydroxylamines”

Answer: Thanks for the reviewer’s suggestion. “N-Arylhydroxylamines” in Lines 11 and 17 was modified to “N-arylhydroxylamines”.

Lines 142, 163, “o-chloronitrobenzene” should be replaced by “o-Chloronitrobenzene”

Answer: Thanks for the reviewer’s suggestion. “o-chloronitrobenzene” in Lines 142 and 163 was modified to “o-Chloronitrobenzene”.

Figure 4: “Ethyl” should be replaced by Ether

Answer: Thanks for the reviewer’s suggestion. “Ethyl” in Figure 4 was modified to “Ether”.

Line 282. “The +R effect of the electron-donating group (9a, 10a) could enrich the electrons of -NHOH, leading to the…” Electron-donating alkyl groups such as methyl (9a) have only positive inductive effect (+I effect). What kind of electronic effect do you mean?

Answer: Thanks for the reviewer’s suggestion. In this sentence, we refer to a positive inductive effect. This sentence was modified to “The electron-donating group (9a, 10a) could enrich the electrons of -NHOH”.

Reviewer 3 Report

The authors present their study of the optimization of conditions for the efficient continuous flow synthesis of N-arylhydroxylamines.  Experimental work is supported by some computational results.  The paper is well written and the conclusions are sound.  The results should be of interest to the greater chemistry community.  I would suggest some minor improvements:

1.  The spectra in the supporting information looks, for the most part, fine.  However, the ortho-chloro product only shows integration for either the NH or OH whereas most of the other spectra show integrations for 1H apiece.  It appears to me that the missing signal is a very broad singlet at ~8.2 ppm and this should be integrated.  Similarly, for the ortho-iodo compound, there is a broad singlet at ~7.5-8.0 ppm that overlaps one of the aryl CH's that appears to correspond to a missing integrated signal as well.  The spectra should be re-integrated to include these signals.

2.  Line 83 and 89:  I believe the percentages provided correspond to product yields.  These should be changed to (0-83% yield) and (39-99% yield) for the sake of clarity.

3.  The full name for DMAP should be provided when it is first mentioned in the text (line 91).

4.  Line 144:  I'm unsure what is meant by "magneton".  Is this a magnetic stir bar?

5.  Line 165:  It is not clear to me what is meant by the phrase "without the influence of external diffusion".  To what does "external diffusion" refer?

6.  Figure 4:  In the legends, diethyl ether is better shortened to be "ether" as opposed to "ethyl"

7.  Table 1:  entries 8 and 9:  the percent yields sum to be greater than 100%!

8.  Figure 8:  It would be easier for the reader if the mechanisms (e.g., HED-1, etc.) were written just below each of the possible scenarios rather than in the figure caption alone.

9.  Figure 9:  The graph should be separate from Figure 9 and have its own descriptive caption.  Also, it is very difficult to visualize what is being displayed on the surfaces.  Can another viewpoint be used instead?

With these changes, I believe this paper worthy for publication.

Author Response

The authors present their study of the optimization of conditions for the efficient continuous flow synthesis of N-arylhydroxylamines.  Experimental work is supported by some computational results.  The paper is well written and the conclusions are sound.  The results should be of interest to the greater chemistry community.  I would suggest some minor improvements:

  1. The spectra in the supporting information looks, for the most part, fine.  However, the ortho-chloro product only shows integration for either the NH or OH whereas most of the other spectra show integrations for 1H apiece.  It appears to me that the missing signal is a very broad singlet at ~8.2 ppm and this should be integrated.  Similarly, for the ortho-iodo compound, there is a broad singlet at ~7.5-8.0 ppm that overlaps one of the aryl CH's that appears to correspond to a missing integrated signal as well.  The spectra should be re-integrated to include these signals.

Answer: Thanks for the reviewer’s suggestion. The spectra of N-(2-chlorophenyl) hydroxylamine (Figure S5) and N-(2-iodophenyl) hydroxylamine (Figure S9) was re-integrated in Supporting Information. Meanwhile, the corresponding compound data has also been modified and marked.

  1. Line 83 and 89:  I believe the percentages provided correspond to product yields.  These should be changed to (0-83% yield) and (39-99% yield) for the sake of clarity.

Answer: Thanks for the reviewer’s suggestion. According to reports in the literature, the percentages in Line 83 provided correspond to product yields, and the percentages in Line 89 provided correspond to product selectivity. (0-83%) and (39-99%) were modified to (0-83% yield) and (39-99% selectivity), respectively.

  1. The full name for DMAP should be provided when it is first mentioned in the text (line 91).

Answer: Thanks for the reviewer’s suggestion. The full name for DMAP was added in Line 91.

  1. Line 144:  I'm unsure what is meant by "magneton".  Is this a magnetic stir bar?

Answer: Thanks for the reviewer’s suggestion. “magneton” was modified to “magnetic stir bar”.

  1. Line 165:  It is not clear to me what is meant by the phrase "without the influence of external diffusion".  To what does "external diffusion" refer?

Answer: Thanks for the reviewer’s suggestion. The "external diffusion" refers to external mass transfer (Org. Process Res. Dev. 2020, 24, 59−66). Under the limitation of external mass transfer, stirring speed has a significant impact on the reaction rate. The experimental data obtained on the premise that the influence of external mass transfer is eliminated can reflect the true rate of the reaction.

  1. Figure 4:  In the legends, diethyl ether is better shortened to be "ether" as opposed to "ethyl"

Answer: Thanks for the reviewer’s suggestion. “Ethyl” in Figure 4 was modified to “Ether”.

  1. Table 1:  entries 8 and 9:  the percent yields sum to be greater than 100%!

Answer: We sincerely thank the reviewer for the constructive suggestion. We are very sorry for not checking again after writing the data. The corresponding modifications have been marked in Table 1.

  1. Figure 8:  It would be easier for the reader if the mechanisms (e.g., HED-1, etc.) were written just below each of the possible scenarios rather than in the figure caption alone.

Answer: Thanks for the reviewer’s suggestion. The mechanisms (e.g., HED-1, etc.) were written below each of the possible scenarios. They were marked in the text.

  1. Figure 9:  The graph should be separate from Figure 9 and have its own descriptive caption.  Also, it is very difficult to visualize what is being displayed on the surfaces.  Can another viewpoint be used instead?

Answer: Thanks for the reviewer’s suggestion. The graph of H2 adsorption on the surface of the catalyst has been adjusted from another viewpoint, and adjusted graph has been added to the SI (Figure S4).